# PieParty: visualizing cells from scRNA-seq data as pie charts

Stefan Kurtenbach[1], James J Dollar[2], Anthony M Cruz[2], Michael A Durante[2], Christina L Decatur[2], J William Harbour[1]

**Single-cell RNA sequencing (scRNA-seq) has been a transformative technology in many research fields. Dimensional reduction techniques such as UMAP and tSNE are used to visualize scRNA-seq data in two or three dimensions for cells to be clustered in biologically meaningful ways. Subsequently, gene expression is frequently mapped onto these plots to show the distribution of gene expression across the plots, for instance to distinguish cell types. However, plotting each cell with only a single color leads to repetitive and unintuitive representations. Here, we present Pie-Party, which allows scRNA-seq data to be plotted such that every cell is represented as a pie chart, and every slice in the pie charts corresponds to the gene expression of a single gene. This allows for the simultaneous visualization of the expression of multiple genes and gene networks. The resulting figures are information dense, space efficient, and highly intuitive. PieParty is publicly available on GitHub at https://github.com/harbourlab/PieParty.**

## Introduction

Gene expression data at single-cell resolution has brought a lot of opportunity to gain detailed understanding of heterogenous cell populations, but also many challenges regarding its processing and visualization. Principle component analysis was initially used to reduce dimensions on single-cell RNA sequencing (scRNA-seq) datasets and to plot them in two or three dimensions. t-SNE and UMAP were developed subsequently, offering superior and global resolution, and are currently the most commonly used dimensional reduction techniques for scRNA-seq (1, 2 Preprint). For most applications, gene expression is mapped onto t-SNE or UMAP plots, for instance, to highlight different cell types or cell states. The common and only approach nowadays is to color the respective cell dots in UMAP or tSNE plots, where the intensity of the color correlates with how high the gene is expressed. However, this only allows plotting one gene per cell, which is inefficient, and results in many repetitive illustrations when expression of multiple different marker genes needs to be shown. This not only consumes valuable figure space but is also not intuitive in many cases. Here, we present PieParty, a python script that allows users to display each cell in t-SNE, UMAP, or any other single-cell plots with coordinates, as pie charts. Each pie chart can be used to visualize expression of multiple genes at once, and can be customized using different colors or color palettes. PieParty also offers additional settings for normalization and customized plotting.

## Results

The basic principle of the PieParty visualization is to generate pie charts for every cell in a single cell sequencing plot, like t-SNE and UMAP plots. The user provides a list of genes that they want to visualize, and PieParty will generate pie charts where each slice in a pie chart represents the proportional gene expression of one individual gene in the list. Each gene (slice) can be assigned a unique color, or a color palette can be chosen to auto-assign unique colors for all genes. Choosing a color palette is useful for larger gene lists. As an example, we analyzed scRNA-seq data from human testis and clustered the cells with UMAP (3). The analysis reveals a continuous developmental trajectory, ranging from stem cells to differentiated sperm, in every defined cell stages. Figs 1 and S1A–C shows PieParty plots, where every cell (n = 15,479) is represented by a pie chart, with 145 different differentiation markers plotted per pie chart. Each differentiation marker is automatically assigned a unique color on a color map, sorted from early markers in dark violet to markers of differentiation in yellow. The resulting plot shows that even plotting a large number of genes in pie charts still yields a very informative and intuitive UMAP plot, showing that the expression of these differentiation markers forms a continuum along the differentiation axis from stem cells to differentiated sperm cells.

Besides merely plotting a single list of genes, PieParty also allows for more complex visualizations. As an example, we used scRNA-seq data from uveal melanoma (UM) tumors to demonstrate a more complex application (4). There are two main classes of UM tumors, class 1, which rarely metastasize, and class 2, which frequently metastasize and have a high mortality rate. Eight biomarkers (*EIF1B*, *FXR1*, *ID2*, *LMCD1*, *LTA4H*,

---

[1]Interdisciplinary Stem Cell Institute, Miami, FL, USA   [2]Bascom Palmer Eye Institute, Sylvester Comprehensive Cancer Center, University of Miami Miller School of Medicine, Miami, FL, USA

Correspondence: stefan.kurtenbach@med.miami.edu

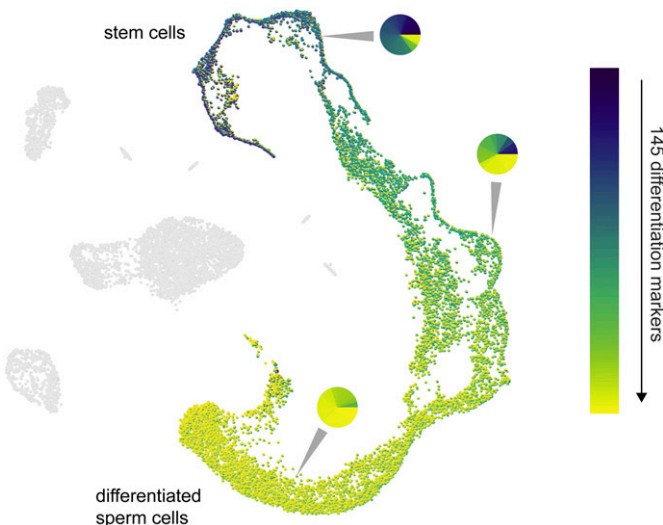

**Figure 1. PieParty plot depicting single cell RNA sequencing data from 5,479 testis cells.**
145 differentiation markers were sorted from early (blue) to late (yellow) and auto-assigned with a color.

*MTUS1*, *ROBO1*, and *SATB1*) are expressed in class 1 tumors, and four biomarker genes are up-regulated in the high-risk class 2 tumors (*CDH1*, *ECM1*, *HTR2B*, and *RRAB31*). The classical way of visualizing this is by showing 12 individual UMAP plots. Fig S2 shows all 12 genes plotted separately, which is very space consuming, and while still possible for 12 genes, not immediately comprehensible. For these and other cases, it is further useful to plot class 1 and class 2 tumor cells separately, which is depicted in Figs S3 and S4. However, with 24 total plots, this consumes even more precious space within a figure using the traditional way. PieParty offers several options to visualize this complex relationship of

class 1 and class 2 genes. One consideration to make is that plotting two or more gene lists with different number of genes, in this case, eight genes for class 1 versus four genes for class 2, comes with a bias which needs to be corrected for. If one gene list is twice the size of the second, but every gene is expressed in the same amount, the pie would be colored more with colors of the longer gene list. Hence, PieParty allows normalization for that fact and weighs gene sets equally by applying a normalization factor to account for the difference in number of genes in each gene list. The normalized expression value ($X_{norm}$) is calculated by multiplication of the expression value ($X$) with the number of total genes in all lists ($N_{total}$), divided by the number of genes in the list of the respective gene ($N_{gene\_list}$), with $X_{norm} = \frac{X * N_{total}}{N_{gene\_list}}$. This normalization technique was applied for all following plots, and the color intensity of the individual pie slices was set to correlate with gene expression, in addition to the size of the pies itself. Fig 2A depicts the tSNE plots of 11 UM tumor samples (three class 1, eight class 2, n = 59,915 cells), plotted with PieParty and a simple blue/red color scheme. Immune cell clusters are indicated, the other cells depicted are tumor cells, with the exception of very small clusters of other cell types (4). Class 1 and class 2 tumor cells were plotted individually, as well as combined in one big plot. It is immediately evident that the tumor cell markers are not only found in tumor cells, but also in immune cells. Class 2 tumor markers are predominantly enriched in macrophages and monocytes, whereas class 1 tumor markers are present in other immune cells including, lymphocytes, NK cells, and T Cells. As can be seen in this example, adding the layer of different color intensities allows comparison of gene expression levels. For instance, some cells appear dark blue, while others are entirely light blue. This shows that genes from the same group are predominantly expressed in those cells, however, with different expression levels. The information density can be further increased by assigning individual colors to each gene, which PieParty can do automatically. Fig 2B shows the same dataset with color palettes applied automatically to the two gene lists. Especially interesting is the fact that in this scRNA-seq dataset some class 1 tumor markers are

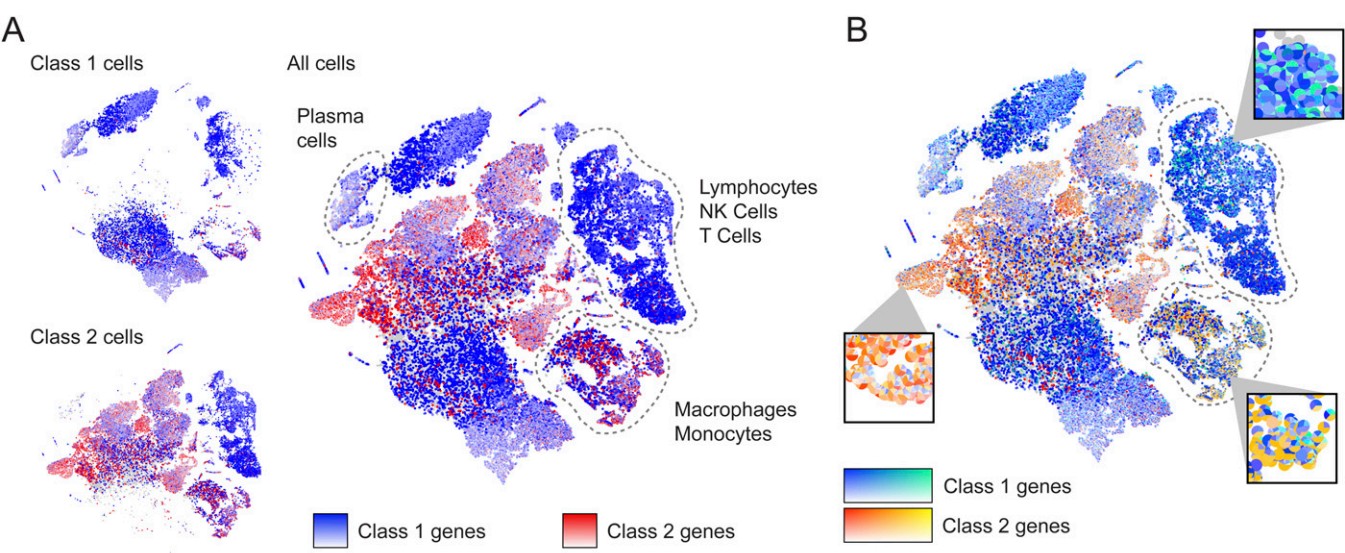

**Figure 2. PieParty representation of 59,915 uveal melanoma (UM) cells. (A)** Class 1 and Class 2 tumor cells plotted separately, with class 1 associated genes in blue, and class 2 associated genes in red. Immune cell clusters are indicated, the other cells are mainly tumor cells. The big plot on the right side of (B) combines all class 1 and class 2 cells ("All cells"). **(B)** Same plot as in (A), but with auto-assigned color palettes, giving each gene a unique color. Color intensity of each slice correlates with gene expression, in addition to slice size.

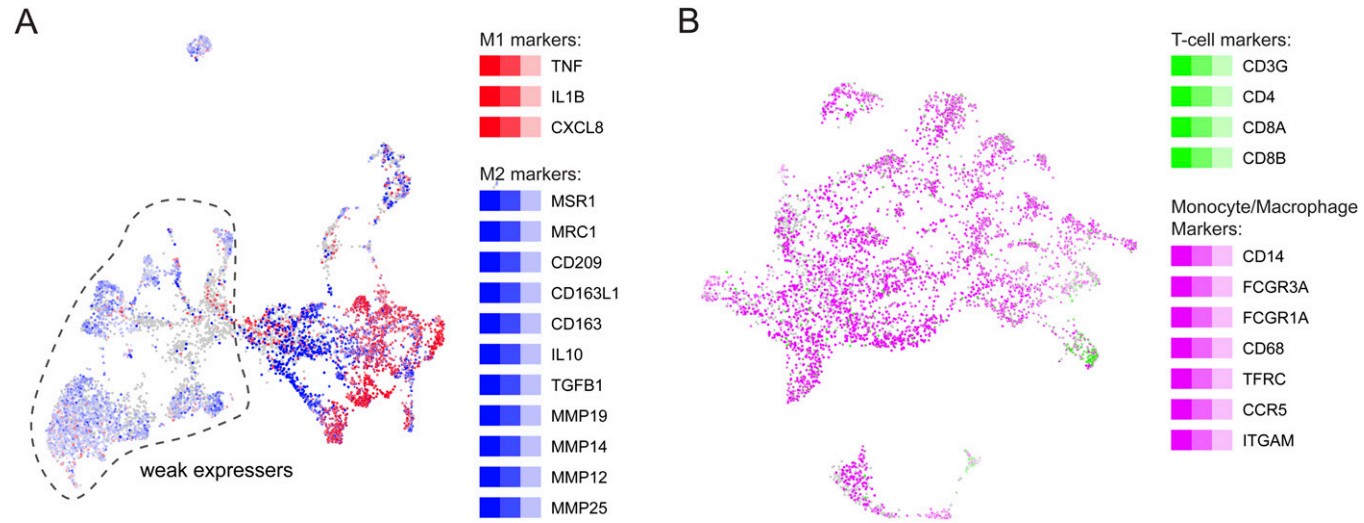

**Figure 3. PieParty plots of immune cell populations of uveal melanoma tumors.**
(A) M1 (red) and M2 (blue) macrophage markers are plotted to distinguish M1 and M2 macrophages. (B) Use of PieParty to plot rare cell populations distinguished by multiple markers. T-cell markers (green) are combined to highlight this rare cell population and distinguish them from monocytes and macrophages. Color intensities correlate with gene expression levels.

expressed in some class 2 immune cells, including ID2, and to some extent SATB1, and EIF1B. Whereas all this information is technically visible when all genes are plotted individually in 24 plots (see Figs S3 and S4), the PieParty plots are intuitively readable and visualize these relationships in a space efficient way. All labels indicating which gene was assigned which colors are generated automatically by PieParty. This visualization shows that for class 2 tumors the biomarkers expressed by immune cells are different from the genes expressed by tumor cells, whereas for class 1 tumors there is overlap. Together, visualizing this dataset with PieParty, proportional expression of biomarkers, as well as expression amount for each gene are all assessable in one single plot.

Another field of application for PieParty is distinguishing cell types and identifying rare cell populations. As an example, we visualized different immune cells from the UM dataset. Fig 3A shows the macrophage population extracted from the complete dataset (n = 8,048 cells), where three M1 macrophage markers assigned the color red and eleven M2 macrophage markers were assigned blue. As mentioned above, PieParty normalized by gene numbers in the different marker lists, which permits for lists with very different number of genes to be used like in this case. Fig 3A shows the clear visualization of M1 and M2 macrophages on the right side of the plot, strongly expressing the indicated markers. Interestingly, there is an equally big population of cells present that express mainly M2 markers but weakly ("weak expressers"), which cluster to the left of the M1 and M2 cells. This provides a striking and clear visualization of macrophage activation heterogeneity and phenotype diversity within the tumor immune microenvironment. Hence, this visualization allows to distinguish these very similar cell types in one plot in a data-driven way, and display the landscape of the expression data. This approach can also be applied to highlight rare cell populations. As an example, we extracted macrophages, monocytes, and undetermined lymphocytes form the UM scRNA-seq dataset (n = 5,111 cells) (Fig 3B). In this dataset, T cells are a rare cell population, comprising only 1.1% of the total cells. We distinguished T cells by the expression of *CD3G*, *CD4*, *CD8A*, and *CD8B* from monocytes and macrophages, which express CD14, *FCGR3A*, *FCGR1A*, *CD68*, *TFRC*, *CCR5*,

and *ITGAM*. The PieParty plot shows how a rare cell population can be clearly distinguished directly in a data-driven way, in contrast to relying on a manually labeled cell cluster.

Another functionality in PieParty is to plot the average gene expression per cell cluster. Fig 4 depicts the "plot clusters" functionality, which generates one pie chart per cell cluster, with the pie chart size correlating with the number of cells in the respective cluster. This plotting style can greatly simplify complex datasets and allow for an intuitive assessment of the expression of different markers in the clusters.

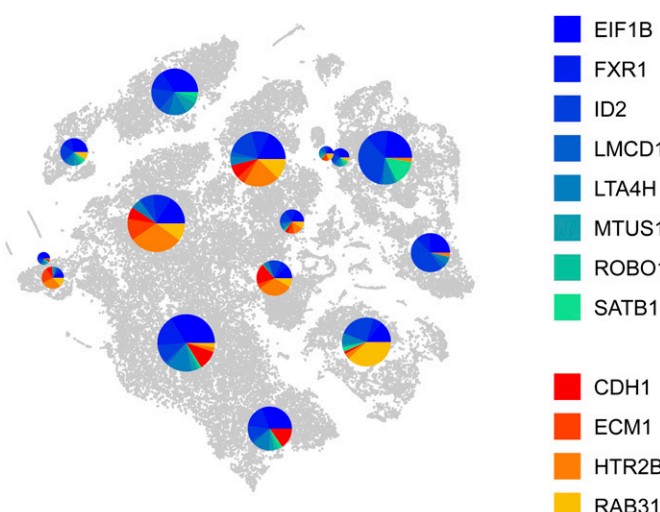

**Figure 4. PieParty cluster plot, using the "big pies" function.**
Each cell cluster is assigned one pie chart depicting the average gene expression in the respective cluster. The chart sizes correlate with the number of cells in the cluster. Class 1 tumor genes were auto-assigned colors from the color map "winter" (blue-green), and class 2 tumor genes were colored with "autumn" (red-yellow).

## Discussion

Here we present PieParty, which allows single cells from scRNA-seq data to be represented as pie charts instead of single-colored dots. PieParty plots are far more information rich, allowing multidimensional single cell sequencing data to be represented in an intuitive and space-efficient way, and for the identification of previously unrecognized transcriptional heterogeneity. We have presented use cases including gene lists with large number of genes to display expression of differentiation markers in sperm differentiation (Fig 1), discern tumor risk classes (Fig 2), characterize immune cell infiltrates (Fig 3A), as well as highlight rare cell populations (Fig 3B). With the public availability of various gene sets, this new visualization technique can be used to visualize various cell types, cell states, pathways, cell cycle, senescence, gene networks, and many more (5). Together, PieParty can provide deeper insights into biology derived from single cell sequencing data by allowing for multidimensional visualization of high-density datasets.

## Materials and Methods

scRNA-seq data for testis and UM are publicly available (3, 4) and were downloaded and analyzed with Seurat (Version 3.2.2) as previously described (4). The pure macrophage population was generated by extracting cells with CD68 expression >1. The macrophages, monocytes, and undetermined lymphocyte population used for rare cell population identification was generated by extracting cell clusters based on cell type classifications generated from the original analysis (4).

PieParty plots were generated using PieParty 1.4 - 1.8, with standard settings. For Fig 1, "lighten colors," "-lc" was set to "False."

## Data Availability

PieParty and example data to test are available on GitHub: https://github.com/harbourlab/PieParty.

## Supplementary Information

## Acknowledgements

This work was supported by the Melanoma Research Foundation Career Development Award (S Kurtenbach) and Established Investigator Award (JW Harbour), National Cancer Institute grant R01 CA125970 (JW Harbour), A Cure in Sight Jack Odell-John Dagres Research Award (S Kurtenbach, JW Harbour), Bankhead-Coley Research Program of the State of Florida (JW Harbour), The Helman Family-Melanoma Research Alliance Team Science Award (JW Harbour), and a generous gift from Dr. Mark J Daily (JW Harbour). This work was supported by the National Institutes of Health (NIH). Center Core grant P30EY014801 and Research to Prevent Blindness- Unrestricted grant (GR004596). The Sylvester Comprehensive Cancer Center also received funding from the National Cancer Institute Core Support grant P30CA240139. The content is solely the responsibility of the authors and does not necessarily represent the official views of the National Institutes of Health.

## Author Contributions

S Kurtenbach: conceptualization, resources, data curation, software, formal analysis, supervision, validation, investigation, visualization, methodology, project administration, and writing—original draft, review, and editing.
JJ Dollar: resources, data curation, visualization, and methodology.
AM Cruz: resources and software.
MA Durante: data curation and writing—original draft.
CL Decatur: conceptualization and resources.
JW Harbour: resources, supervision, funding acquisition, and writing—original draft.

### Conflict of Interest Statement

Dr. JW Harbour is an inventor of intellectual property discussed in this study. He is a paid consultant for Castle Biosciences, licensee of this intellectual property, and he receives royalties from its commercialization.

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
