## [Reviewer comments · Life Science Alliance]

Life Science Alliance

PieParty: Visualizing cells from scRNA-seq data as pie charts

James J. Dollar, Anthony M Cruz, Michael Durante, Christina L Decatur, J William Harbour and Stefan Kurtenbach

DOI: <https://10.26508/lsa.202000986>

Corresponding author(s): Stefan Kurtenbach (University of Miami)

Review Timeline:

Submission Date:	2020-12-10
Editorial Decision:	2021-01-06
Revision Received:	2021-01-25
Editorial Decision:	2021-02-08
Revision Received:	2021-02-12
Accepted:	2021-02-17

Scientific Editor: Shachi Bhatt

Transaction Report:

January 6, 2021

Re: Life Science Alliance manuscript #LSA-2020-00986-T

Dr. Stefan Kurtenbach
Ruhr University Bochum
Cell Physiology
Bochum D-44780

Dear Dr. Kurtenbach,

Thank you for submitting your manuscript entitled "PieParty: Visualizing cells from scRNA-seq data as pie charts" to Life Science Alliance. The manuscript was assessed by expert reviewers, whose comments are appended to this letter.

As you will note from the reviewers' comments, the reviewers were enthusiastic about the tool, but have some concerns about the error message (Rev 1) and interpretation of the pie charts. We thus invite you to submit a revised manuscript that addresses all of the reviewers concerns.

Thank you for this interesting contribution to Life Science Alliance. We are looking forward to receiving your revised manuscript.

Sincerely,

Shachi Bhatt, Ph.D.
Executive Editor
Life Science Alliance

- A letter addressing the reviewers' comments point by point.
- An editable version of the final text (.DOC or .DOCX) is needed for copyediting (no PDFs).
- High-resolution figure, supplementary figure and video files uploaded as individual files: See our detailed guidelines for preparing your production-ready images, <https://www.life-science-alliance.org/authors>
- Summary blurb (enter in submission system): A short text summarizing in a single sentence the study (max. 200 characters including spaces). This text is used in conjunction with the titles of papers, hence should be informative and complementary to the title and running title. It should describe the context and significance of the findings for a general readership; it should be written in the present tense and refer to the work in the third person. Author names should not be mentioned.

B. MANUSCRIPT ORGANIZATION AND FORMATTING:

Reviewer #1 (Comments to the Authors (Required)):

Single cell sequencing produce a huge amount of information so data visualization is a major challenge. Herein the authors have generated a tool that enable the data to be visualized as piecharts which dramatically reduces the space needed in for eg journal publication. The tool is also useful in that many different genes can be shown for co-expression interpretations as well as studies of gene signatures. The authors demonstrate utility of the tool in an appropriate way. We can conclude that this will be a useful addition if it works well. However, upon testing the software tool the following error message was received:

Traceback (most recent call last):
File "bin/PieParty/PieParty.py", line 258, in

```
pie_colors.append(pick_color(colors[file_nr], percentile, lighten))
File "bin/PieParty/PieParty.py", line 39, in pick_color
return(lighten_color(return_color, lighten))
File "bin/PieParty/PieParty.py", line 24, in lighten_color
return pl.colors.to_hex(colors.hls_to_rgb(c[0], 1 - amount * (1 - c[1]), c[2]))
File "/home/.local/lib/python3.7/site-packages/matplotlib/colors.py", line 309, in to_hex
c = to_rgba(c)
File "/home/.local/lib/python3.7/site-packages/matplotlib/colors.py", line 177, in to_rgba
rgba = _to_rgba_no_colorcycle(c, alpha)
File "/home/.local/lib/python3.7/site-packages/matplotlib/colors.py", line 250, in
_to_rgba_no_colorcycle
raise ValueError("RGBA values should be within 0-1 range")
ValueError: RGBA values should be within 0-1 range
```

Can the authors help explaining this error message and how to solve the problem? Which Python module was used when creating the tool?

Reviewer #2 (Comments to the Authors (Required)):

The paper by Kurtenbach et al entitled "PieParty: Visualizing cells from scRNA-seq data as pie charts" describes a novel approach for visualisation of single-cell data (implemented as a Python script). The authors suggest to visualise each cell as a pie-chart showing expression levels of several or multiple genes.

While this is an interesting idea, I have concerns about interpretation of such figures. First, the pie charts show proportional gene expression in each cell. So when we see some areas on a UMAP coloured in blue, while others - in red (e.g. Figure 2), we don't know what is happening at the gene level: is it that 'blue genes' have constant expression levels, while 'red genes' have different expression in different cell types, is it the other way around, or both gene sets have varying gene expression.

Second, I am concerned about data overplotting. Naturally most of the pies are partially covered by others, and we see an incomplete picture. Moreover, if in a certain area the pies are mapped in a way that each next plotted pie covers the bottom part of the ones plotted before, we will see largely the top parts of the pies, and visually it will look like high expression of genes shown on the top of each pie.

Third, not clear whether gene expression is shown on the same scale for all genes or not. E.g. does dark blue on Figure 3a start at the same value for MSR1, MRC1 and all other genes? And more generally, the information on the strength of the gene expression is missing. We can see that the expression level varies, but does it vary from 0.01 to 0.03 or from 0.01 to 4? Would be useful to have expression values labelled on the legend.

Minor points:

Line 54 mentions "145 different differentiation markers", whereas Figure 1 says 150 markers - I think this number doesn't need to be rounded, better show the same for consistency.

Line 76 - "Hence, PieParty allows to normalize for that fact, and weighs gene sets equally by

applying a normalization factor to account for the difference in number of genes in each gene list" - it is hard to understand from this sentence what the normalization approach is exactly - please explain in details, provide a formula.

Line 87 - "Especially interesting is the fact that in this scRNA-seq dataset one class 1 tumor marker is expressed in some class 2 immune cells." - how can the authors tell from the figure, that it was just one gene (if all class 1 genes are shown in blue)?

Line 96 - "the biomarkers expressed by immune cells are different from the genes expressed by tumor cells" - what dots correspond to tumour cells? We see just two groups of immune cells annotated on Figure 2a, but we don't know what the other cells are.

Line 105-107 - "Figure 3a shows the clear visualization of M1 and M2 macrophages on the right side of the plot, strongly expressing the indicated markers. Interestingly, there is an equally big population of cells present that express mainly M2 markers but weakly" - I suggest to label the populations on the plot to avoid ambiguity.

Lastly, I'd suggest to upload example input files to GitHub and also to mention an estimate of time the script runs.

Reviewer #1 (Comments to the Authors (Required)):

Single cell sequencing produce a huge amount of information so data visualization is a major challenge. Herein the authors have generated a tool that enable the data to be visualized as piecharts which dramatically reduces the space needed in for eg journal publication. The tool is also useful in that many different genes can be shown for co-expression interpretations as well as studies of gene signatures. The authors demonstrate utility of the tool in an appropriate way. We can conclude that this will be a useful addition if it works well. However, upon testing the software tool the following error message was received:

Traceback (most recent call last):

```
File "bin/PieParty/PieParty.py", line 258, in <module>
  pie_colors.append(pick_color(colors[file_nr], percentile, lighten))
File "bin/PieParty/PieParty.py", line 39, in pick_color
  return(lighten_color(return_color, lighten))
File "bin/PieParty/PieParty.py", line 24, in lighten_color
  return pl.colors.to_hex(colors.sys.hls_to_rgb(c[0], 1 - amount * (1 - c[1]), c[2]))
File "/home/.local/lib/python3.7/site-packages/matplotlib/colors.py", line 309, in to_hex
  c = to_rgba(c)
File "/home/.local/lib/python3.7/site-packages/matplotlib/colors.py", line 177, in to_rgba
  rgba = _to_rgba_no_colorcycle(c, alpha)
File "/home/.local/lib/python3.7/site-packages/matplotlib/colors.py", line 250, in
  _to_rgba_no_colorcycle
  raise ValueError("RGBA values should be within 0-1 range")
ValueError: RGBA values should be within 0-1 range
```

Can the authors help explaining this error message and how to solve the problem? Which Python module was used when creating the tool?

We have tested this tool with various datasets and do not encounter this error. It is hard to troubleshoot this with only the error message unfortunately. As request of the other reviewer, we have now included an example dataset on the GitHub page, which can be used to troubleshoot this issue. In addition, we welcome you to use the “raise issue” function on GitHub, and I will be happy to help troubleshoot and get this running for you!

Reviewer #2 (Comments to the Authors (Required)):

The paper by Kurtenbach et al entitled "PieParty: Visualizing cells from scRNA-seq data as pie charts" describes a novel approach for visualisation of single-cell data (implemented as a Python script). The authors suggest to visualise each cell as a pie-chart showing expression levels of several or multiple genes.

While this is an interesting idea, I have concerns about interpretation of such figures. First, the pie charts show proportional gene expression in each cell. So when we see some areas on a UMAP coloured in blue, while others - in red (e.g. Figure 2), we don't know what is happening

at the gene level: is it that 'blue genes' have constant expression levels, while 'red genes' have different expression in different cell types, is it the other way around, or both gene sets have varying gene expression.

PieParty not only makes the slices of pie charts proportional to the gene expression in each cell, but also by default changes the color intensity in respect to the expression in the dataset. As can be seen in the mentioned Figure 2, there are cells that are entirely dark blue, or entirely light blue. This reflects that the same group of genes is expressed in both, however, in different amounts. This allows the user to compare expression values directly. We realize that our initial description was a little sparse, and added a clearer description to the manuscript and the GitHub page, highlighting this difference.

Second, I am concerned about data overplotting. Naturally most of the pies are partially covered by others, and we see an incomplete picture. Moreover, if in a certain area the pies are mapped in a way that each next plotted pie covers the bottom part of the ones plotted before, we will see largely the top parts of the pies, and visually it will look like high expression of genes shown on the top of each pie.

PieParty uses any coordinate file e.g. from UMAP or tSNE representations as inputs. While the pie charts are slightly larger than regular dots for cell representations in normal plots, the size difference is not as big as that overlapping cells causes significant more issues as in any regular single cell plot. It is upon the user to adjust clustering parameters for a desired outcome and sufficient cell separation, whether PieParty is used or not. For the datasets we have investigated, the stochastic nature of cell plotting and large cell numbers of single cell datasets in general prevented regions where only bottom parts of pie charts are visualized to accumulate. A proof of principle can be seen in Figure 1, where large amounts of overlapping cells were plotted, and even without the setting to correlate color saturation with expression levels, gives a very clear result.

Third, not clear whether gene expression is shown on the same scale for all genes or not. E.g. does dark blue on Figure 3a start at the same value for MSR1, MRC1 and all other genes? And more generally, the information on the strength of the gene expression is missing. We can see that the expression level varies, but does it vary from 0.01 to 0.03 or from 0.01 to 4? Would be useful to have expression values labelled on the legend.

Indeed, the same color means same expression value in this case. We added a better description to the manuscript and the GitHub page.

Minor points:

Line 54 mentions "145 different differentiation markers", whereas Figure 1 says 150 markers - I think this number doesn't need to be rounded, better show the same for consistency.

Thank you for finding this discrepancy. We changed the Figure legend to 145.

Line 76 - "Hence, PieParty allows to normalize for that fact, and weighs gene sets equally by applying a normalization factor to account for the difference in number of genes in each gene list" - it is hard to understand from this sentence what the normalization approach is exactly - please explain in details, provide a formula.

We have now added a detailed description and formula to the manuscript.

Line 87 - "Especially interesting is the fact that in this scRNA-seq dataset one class 1 tumor marker is expressed in some class 2 immune cells." - how can the authors tell from the figure, that it was just one gene (if all class 1 genes are shown in blue)?

This sentence should have been after Figure 2b was mentioned, in which we had used different colors for all genes. Besides this, while ID2 was the predominant class 1 marker found in class 2 immune cells, there indeed are others which are visible in Figure 2b. We expanded on this in the manuscript. Thank you for pointing this out!

Line 96 - "the biomarkers expressed by immune cells are different from the genes expressed by tumor cells" - what dots correspond to tumour cells? We see just two groups of immune cells annotated on Figure 2a, but we don't know what the other cells are.

The other cells are predominantly tumor cells, with only very small clusters from other cell types. One exception are plasma cells, which have a larger cluster, and we now have annotated this one in addition. We also added this description to the manuscript – which was lacking – and a reference to our single cell paper where the small clusters are more comprehensively annotated for the readers interested.

Line 105-107 - "Figure 3a shows the clear visualization of M1 and M2 macrophages on the right side of the plot, strongly expressing the indicated markers. Interestingly, there is an equally big population of cells present that express mainly M2 markers but weakly" - I suggest to label the populations on the plot to avoid ambiguity.

We now highlighted the “weak expressers” cluster in the figure.

Lastly, I'd suggest to upload example input files to GitHub and also to mention an estimate of time the script runs.

Thank you for this excellent suggestion! We have added example input files and description on GitHub for the users to test, which has already been very helpful for some users in the last couple days. We also added a time estimation for it.

February 8, 2021

RE: Life Science Alliance Manuscript #LSA-2020-00986-TR

Author information redacted

Dear Dr. Kurtenbach,

Thank you for submitting your revised manuscript entitled "PieParty: Visualizing cells from scRNA-seq data as pie charts". We would be happy to publish your paper in Life Science Alliance pending final revisions necessary to meet our formatting guidelines.

Along with the points listed below, please also attend to the following,

- please consult our manuscript preparation guidelines <https://www.life-science-alliance.org/manuscript-prep> and make sure your manuscript sections are in the correct order;
- please be sure that all Authors are mentioned in the Author Contributions section in your main manuscript text
- please add a conflict of interest statement to your main manuscript text
- please upload your main and supplementary figures as single files
- please revise the legend for figure S1 so that the panels are introduced
- please add callouts for Figure S1A, B, C to your main manuscript text
- there is a callout for Fig1A, although the actual figure doesn't have it nor its legend. Please revise
- please upload your main manuscript text as an editable doc file
- we can see that you have shared the github link for PieParty in the Abstract. We also encourage you to add it under a 'Data Availability' section in the manuscript text

A. FINAL FILES:

B. MANUSCRIPT ORGANIZATION AND FORMATTING:

Sincerely,

Shachi Bhatt, Ph.D.
Executive Editor
Life Science Alliance
<https://www.lsjournal.org/>
Tweet @SciBhatt @LSAJournal

Interested in an editorial career? EMBO Solutions is hiring a Scientific Editor to join the international Life Science Alliance team. Find out more here - https://www.embo.org/documents/jobs/Vacancy_Notice_Scientific_editor_LSA.pdf

Reviewer #1 (Comments to the Authors (Required)):

The authors did not explain the error message we had encountered but provided sample data and explanation how to get help so this might clarify.
I think the study can be published in its current form.

Reviewer #2 (Comments to the Authors (Required)):

My concerns have been addressed in this revision

February 17, 2021

RE: Life Science Alliance Manuscript #LSA-2020-00986-TRR

Author information redacted

Dear Dr. Kurtenbach,

Thank you for submitting your Research Article entitled "PieParty: Visualizing cells from scRNA-seq data as pie charts". It is a pleasure to let you know that your manuscript is now accepted for publication in Life Science Alliance. Congratulations on this interesting work.

DISTRIBUTION OF MATERIALS:

Again, congratulations on a very nice paper. I hope you found the review process to be constructive and are pleased with how the manuscript was handled editorially. We look forward to future exciting submissions from your lab.

Sincerely,

Shachi Bhatt, Ph.D.
Executive Editor
Life Science Alliance
<https://www.lsjournal.org/>

Interested in an editorial career? EMBO Solutions is hiring a Scientific Editor to join the international Life Science Alliance team. Find out more here -

https://www.embo.org/documents/jobs/Vacancy_Notice_Scientific_editor_LSA.pdf